# Estimating the global cost of vision impairment and its major causes: protocol for a systematic review

Ana Patricia Marques ,[1] Jacqueline Ramke ,[1,2] John Cairns,[3] Thomas Butt,[4] Justine H Zhang ,[1] Hannah B Faal,[5] Hugh Taylor,[6] Iain Jones,[7] Nathan Congdon,[8,9] Andrew Bastawrous,[1] Tasanee Braithwaite,[1,10] Marty Jovic,[11] Serge Resnikoff ,[12] Allyala Nandakumar,[13] Peng Tee Khaw,[14] Rupert R A Bourne,[15] Iris Gordon,[1] Kevin Frick,[16] Matthew J Burton[1,10]

For numbered affiliations see end of article.

**Correspondence to**
Ana Patricia Marques;
Patricia.Marques@lshtm.ac.uk

## ABSTRACT

**Introduction** Vision impairment (VI) places a burden on individuals, health systems and society in general. In order to support the case for investing in eye health services, an updated cost of illness study that measures the global impact of VI is necessary. To perform such a study, a systematic review of the literature is needed. Here we outline the protocol for a systematic review to describe and summarise the costs associated with VI and its major causes.

**Methods and analysis** We will systematically search in Medline (Ovid) and the Centre for Reviews and Dissemination database which includes the National Health Service Economics Evaluation Database. No language or geographical restriction will be applied. Additional literature will be identified by reviewing the references in the included studies and by contacting field experts. Grey literature will be considered. The review will include any study published from 1 January 2000 to November 2019 that provides information about costs of illness, burden of disease and/or loss of well-being in participants with VI due to an unspecified cause or due to one of the seven leading causes globally.

Two reviewers will independently screen studies and extract relevant data from included studies. Methodological quality of economic studies will be assessed based on the *British Medical Journal* checklist for economic submissions adapted to costs of illness studies. This protocol has been prepared following the Preferred Reporting Items for Systematic Reviews and Meta-Analysis protocols and has been published prospectively in Open Science Framework.

**Ethics and dissemination** Formal ethical approval is not required, as primary data will not be collected in this review. The findings of this study will be disseminated through peer-reviewed publications, stakeholder meetings and inclusion in the ongoing Lancet Global Health Commission on Global Eye Health.

**Registration details** https://osf.io/9au3w (DOI 10.17605/OSF.IO/6F8VM).

## Strengths and limitations of this study

► This protocol adheres to the Preferred Reporting Items for Systematic Reviews and Meta-Analysis protocols and has been published prospectively in Open Science Framework.
► This systematic review will search various databases extensively and will include studies published from 1 January 2000 to November 2019 without any language or geographical restriction.
► All included studies will be appraised using the *British Medical Journal* checklist for economic submissions adapted for the cost of illness studies.
► Synthesis of findings will be difficult as resource use (including diagnostic procedures and treatment options) and costs will likely vary between countries, over time and according to which cause(s) of vision loss is reported—in lieu of synthesis, we will summarise the range and quality of available evidence, and the subsequent gaps where evidence should be produced and improved.
► Due to the expected heterogeneity in study methods, it is unlikely that a meta-analysis will be conducted.

were blind (visual acuity worse than 3/60 in the better eye) and 216.6 million (80% uncertainty interval 98.5–359.1) were moderately or severely visually impaired (visual acuity better than 3/60 but worse than 6/18 in the better eye).[1] In 2015, 87% of blindness and 75% of moderate and severe VI was due to seven causes—uncorrected refractive error, cataract, glaucoma, age-related macular degeneration (AMD), diabetic retinopathy, corneal opacity and trachoma.[2]

VI—being the combination of blindness and moderate and severe VI—is associated with a range of consequences including difficulties performing activities of daily living,[3–5] reduced mobility,[6] higher risk of depression,[7 8] reduced educational outcomes,[9] impaired workplace productivity,[10] decreased

## BACKGROUND

Vision impairment (VI) is a major public health issue. In 2015 an estimated 36 million people (80% uncertainty interval 12.9–65.4)

quality of life,[11] increased risk of falls,[12] higher levels of dependency,[3] increased need for informal and formal care,[13–15] and an increased need for healthcare.[16–18] All of these lead to an economic burden for individuals, health systems and society. VI may occur at any age due to genetic, acquired or trauma-related causes. However, the prevalence of VI increases with age in all world regions.

In 2010, the only global estimate for the cost of VI conducted to date reported a cost of US$2954 billion,[19] with direct costs of US$2302 billion and informal care costs of US$246 million.[19] This analysis included productivity losses for high-income countries only, and in 2010 these were estimated to be US$168.3 billion.[19]

Another estimate of productivity losses due to VI has been reported in a study that used data from nine countries from high-income, middle-income and low-income countries and three different analysis approaches.[20] The most conservative of these approaches estimated that productivity losses due to VI in 2011 ranged from US$0.1 billion in Honduras to US$7.8 billion in USA.[20] The authors concluded that although VI occurs more frequently in low-income and middle-income countries, the economic burden is still substantial in high-income countries, such as USA and Japan.[20] Further, the full cost of VI is conceivably much higher if direct and informal care costs were included in estimates.

In order to make a case for investment and to develop plans to alleviate the burden of VI, an updated cost of illness study measuring the global impact from an economic and societal perspective is necessary.

Cost of illness studies measure the economic burden of a disease or condition on the overall population.[21 22] They are descriptive and analytic studies that estimate all direct healthcare costs, productivity losses and intangible costs of a disease or illness.[23] They are conducted to advise healthcare planners about the size of a problem in a population, to update and support policy and financing decisions and to inform full economic evaluation studies, namely cost-effectiveness and cost–benefit analyses.[24 25] Cost of illness studies do not compare alternative interventions and as such are considered partial economic evaluation studies.[26 27]

To perform a global cost of illness study, all available data must be identified and collated in a structured way. In 2012, a systematic review was conducted to inform a cost of illness study on VI and main causes of VI in high-income countries and a total of 22 studies were identified that reported direct and/or indirect costs related to VI.[28] Since 2012, new treatments (eg, anti-vascular endothelial growth factor (anti-VEGF) therapy) and technologies (eg, ocular imaging) have emerged. These are expected to increase direct costs and, if effective, improve outcomes.

A new systematic review is now required for three reasons. First, the search will be extended to include low-income and middle-income as well as high-income countries to allow comprehensive global estimates. Second, we will expand the search to include the seven major causes of VI identified in the latest global prevalence estimates—cataract, uncorrected refractive error, diabetic retinopathy, glaucoma, AMD, corneal opacity and trachoma.[2] Finally, a new systematic review will capture studies on new treatments, such as anti-VEGF treatment, which may result in both substantial costs and savings, and are thus likely to affect the societal cost of VI.

## PURPOSE

The aim of this systematic review is to describe and summarise the costs associated with VI and its major causes.

## METHODS AND ANALYSIS

The protocol is reported in accordance with the Preferred Reporting Items for Systematic Reviews and Meta-Analysis protocols (PRISMA-P) checklist[29 30] (online supplementary annex 1) and has been registered previously in Open Science Framework.

### Search

Literature searches will be performed in Medline (Ovid) and the Centre for Reviews and Dissemination database which includes the National Health Service Economics Evaluation Database, Database of Abstracts of Reviews of Effects and the Health Technology Assessment database. Searches will be run to identify studies published from 1 January 2000 to November 2019, and no language or geographical restrictions will be applied. The search strategy is provided in online supplementary annex 2.

The reference lists of included articles will be reviewed for additional relevant articles. Field experts, including health economists and eye care researchers who have conducted economic evaluation in eye care will be contacted to identify further potentially relevant studies and reports in the grey literature. These individuals will be identified from the authorship of the identified articles and snowballing via recommendations from Commissioners in the Lancet Commission on Global Eye Health.

### Criteria

Studies will be included if they:
► are partial economic evaluation studies such as cost of illness studies, burden of illness/diseases and full economic evaluation studies such as cost-effectiveness and cost–benefit studies published since 1 January 2000; and
► report in the results section a monetary estimate of the direct and/or indirect and/or productivity and/or informal care costs associated with persons with VI from an unspecified cause or due to one of the seven leading causes of vision loss globally (ie, cataract, uncorrected refractive error, diabetic retinopathy, glaucoma, AMD, corneal opacity and trachoma); and/or
► report at least one of:
 – undiscounted or discounted cost or benefit results; and/or

– an estimate of the impact of VI on labour market outcomes (eg, employment chances, labour income, wages and lost work days), informal care (eg, number of caregiver hours) or in terms of well-being (eg, Quality Adjusted Life Years (QALYs), Disability Adjusted Life Years (DALYs).

Studies will be excluded if they:

► only report incremental costs, net costs, incremental benefits or net benefits, incremental cost-effectiveness ratio, incremental cost–benefit ratios without also reporting actual costs; or

► report costs and benefits related to specific eye diseases that are not one of the seven leading causes of vision loss globally; or

► report costs of services for people with one of the major causes of VI (eg, screening for everyone with diabetic retinopathy, providing medication for everyone with glaucoma) without specifically reporting the costs to deliver the service to people with VI. The exceptions will be studies reporting costs of services to treat cataract and refractive error—these will be included regardless of the vision status of participants, as they tend to be single (for cataract) or irregular (for refractive error) interventions that correct the VI, compared with the services required for the other causes; or

► are reviews of existing economic studies related to VI; or

► report an economic model based on other studies, but do not report any primary costs data.

Inclusion criteria are summarised and complemented with PICOS details in table 1.

### Methodological features of cost of illness studies

Cost of illness studies follow two different epidemiological approaches: prevalence-based or incidence-based approaches.[21 31] Prevalence-based studies estimate costs associated with prevalent cases over a given period of time (usually 1 year), while incident-based studies estimate costs accrued over a lifetime following the onset of the illness or loss of health state.

Cost of illness studies can be conducted from various perspectives, including societal, governmental, healthcare system, payer, healthcare provider and patient.[21] The analysis approach varies with the chosen perspective and may include direct costs, productivity costs, informal care costs and intangible costs.[25 32]

*Direct costs* may include direct medical and non-medical costs associated with inpatient and outpatient care and all the resources used for diagnosis and treatment of eye disease and its sequelae, long-term care and nursing home costs, community care and paid assistance provided by professionals, costs related to vision aids and devices and home modifications and transportation costs to access services. *Productivity costs* (formerly called indirect costs) may include absenteeism, presenteeism, reduced workforce participation and lost productivity due to premature mortality. *Informal care* may include hours spent by caregivers and/or a monetary estimate of the hours spent in care. *Intangible costs* are captured through QALYs and DALYs. *Transfer payments* such as social welfare payments made for distributional purposes. *Deadweight losses* namely the cost to society of administering certain transfer payments, such as social welfare payments.

Resource consumption estimates depend largely on the characteristics of the available data[21 32] and are usually categorised as top-down ('population-level') or bottom-up ('person-based').[21] Top-down methods use aggregate expenditures by cost component while the bottom up method assigns costs to individuals with a specific disease or condition.

### Selection of sources of evidence

All titles and abstracts will be screened by two investigators independently (APM and one of JR, JZ, ThB) using Covidence systematic review software (Veritas Health Innovation, Melbourne, Australia. Available at www.covidence.org). After completing the screening process, full texts will be assessed by two investigators independently to establish eligibility for inclusion into the study. Since formal international guidelines for quality assessment of economic studies are lacking,[33] all included studies will be appraised by two investigators independently using the *British Medical Journal* checklist[34] for economic submissions adapted for cost of illness studies.[25] Each quality criteria will be scored as one

| Table 1 | Summary of the PICOS elements for the systematic review |
|---|---|
| *Participants* | Participants with VI from an unspecified cause or due to one of the leading causes of VI globally (ie, cataract, uncorrected refractive error, diabetic retinopathy, glaucoma, AMD, corneal opacity and trachoma). |
| *Interventions* | Any report that provides information about costs of illness, burden of diseases and/or loss of well-being in participants with VI or eye disease potentially leading to VI. |
| *Comparators* | Not relevant |
| *Outcomes* | Direct costs, indirect costs, productivity losses, informal care and intangible costs (eg, Quality Adjusted Life Years, Disability Adjusted Life Years), transfer payments and deadweight losses. |
| *Study Design* | Partial economic evaluation studies such as cost of illness studies, burden of illness/diseases and full economic evaluation studies, such as cost-effectiveness and cost–benefit studies. Model-based economic evaluation studies not reporting any primary cost data or based on reviews of existing economic studies will be excluded. |

AMD, age-related macular degeneration.

of 'yes', 'no', 'partial' or 'not applicable'. We will follow the approach used several times previously to identify the methodological strengths and weakness of the included studies[32 35 36]—equal weight will be assigned to each item of the checklist and the final score will be equal to the sum of the 10 individual items. Any conflict in relation to screening and appraisal will be discussed between the two investigators, and resolved with a third investigator if necessary. A PRISMA flow diagram will be completed to summarise the study selection process.

## Data extraction characteristics

The following information will be extracted from the included studies:

► Country or countries of study.
► Study period.
► Study size (eg, population-based studies or sampled-based studies).
► Age range of participants.
► Study design (eg, cost of illness, burden of illness/diseases, cost-effectiveness or cost–benefit studies).
► Epidemiological approach (eg, incidence-based or prevalence-based).
► Perspective of analysis (eg, societal, government, healthcare system, payer, healthcare provider or patient).
► Main data sources (eg, published expenditures report, administrative database, population survey, patient clinical records, patient diaries, specially designed questionnaires, published literature).
► Method of resource quantification (eg, top-down or bottom-up).
► VI definition and VI severity (eg, blind, moderate or severe VI).
► Cause of VI (and definition).
► Disease stage.
► Currency in which costs are reported.
► Cost components (eg, direct costs, productivity costs, informal care costs).
► Loss of well-being measures (eg, intangible costs measured with QALYs, DALYs, years of sight loss).
► Analysis of uncertainty (eg, type of uncertainty analysed (parameter uncertainty, methodological uncertainty or modelling uncertainty), choice of parameters included in sensitivity analysis, univariate sensitivity analysis, probabilistic sensitive analysis).
► Discounting methods (eg, discount rate applied and justification).

If the study perspective or the epidemiological approach is not clearly specified in the studies, two investigators will assign a category for it by consensus.

## Synthesis of results

Selected studies will be characterised in terms of country of origin, epidemiological approach, perspective of analysis, study design, study size, methods of resource quantification and methods to deal with uncertainty.

We will describe the main reported cost categories and the general assumptions used to estimate costs. We will take four steps to prepare study results for comparison:

1. We will categorise studies either as 'general' studies that reported costs for people with blindness or VI or 'condition'-specific studies that reported costs for people with one of the seven specified causes of vision loss.
2. If costs per patient per year are not reported for national or global estimates studies, these will be calculated for studies where sufficient information is provided.
3. Costs will be inflated to 2018 values (or to the most recent available year) using country-specific gross domestic product deflators.[37]
4. Costs will be converted to USD purchasing power parities (PPP)[38] to equalise the purchasing power of different currencies.

Time transformations will adjust for inflation costs reported in the same country but in different years. Conversion to USD PPP conversion will adjust for the same price level costs estimates reported in different countries and different currencies. This cost transformation will convert all reported costs to the same year (2018), same currency and same purchasing power (USD PPP).

Due to anticipated heterogeneity in the cost data, studies will be stratified and presented by the four different cost components (ie, direct costs, productivity losses, informal care and intangible costs), with a clear explanation of what has been included in each of the four cost components. A table summarising which items are included in the four major cost components will be reported to summarise the similarities and differences between studies. Cost data will also be stratified by severity of VI when this information is available. Since this systematic review aims to collect data to assist a future global economic estimate for VI and its major causes, the transformed cost per patient per year stratified by cost components will be aggregated by Global Burden of Diseases regions and super regions. Descriptive measures will be calculated to report the cost per patient per year for each region and super region (eg, mean, SD, minimum and maximum).

We will describe the main reported loss of well-being measures and its general assumptions. Loss of well-being measures will be summarised in their natural units (eg, QALYS and DALYS) rather than reported in their monetised value since there is no consensus on assigning a monetary value to health outcomes[21 26 39] and because there is no common acceptable value across countries.

Due to the expected heterogeneity in study design, definitions of costs/loss of well-being,[40] it is unlikely that a meta-analysis will be conducted.

## ETHICS AND DISSEMINATION

Formal ethical approval is not required, as primary data will not be collected in this review. The findings of this study will be disseminated through a peer-reviewed publication, stakeholder meetings and inclusion in the ongoing Lancet Global Health Commission on Global Eye Health.[41]

## PATIENT AND PUBLIC INVOLVEMENT

Patients and the public were not involved in the design of this systemic review protocol.

### Author affiliations
[1]International Centre for Eye Health, London School of Hygiene and Tropical Medicine, London, UK
[2]School of Optometry and Vision Science, University of Auckland, Auckland, New Zealand
[3]Department of Health Services Research and Policy, London School of Hygiene and Tropical Medicine, London, UK
[4]UCL Institute of Ophthalmology, University College London, London, UK
[5]Department of Ophthalmology, University of Calabar, Calabar, Nigeria
[6]University of Melbourne School of Population and Global Health, Melbourne, Victoria, Australia
[7]Sightsavers, Haywards Heath, UK
[8]Centre for Public Health, Queen's University, Belfast, UK
[9]Sun Yat-Sen University Zhongshan Ophthalmic Center, Guangzhou, Guangdong, China
[10]Moorfields Eye Hospital, London, UK
[11]PricewaterhouseCoopers, Sydney, New South Wales, Australia
[12]Brien Holden Vision Institute, University of New South Wales, Sydney, New South Wales, Australia
[13]Brandeis University Heller School for Social Policy and Management, Waltham, Massachusetts, USA
[14]NIHR Biomedical Research Centre at Moorfields Eye Hospital and UCL Institute of Ophthalmology, London, UK
[15]Vision and Eye Research Unit, Anglia Ruskin University, Cambridge, UK
[16]Johns Hopkins University Carey Business School - Baltimore Campus, Baltimore, Maryland, USA

**Contributors** APM, JC, JR and MJB conceived the idea for the review. APM, JR and ThB drafted and revised the protocol with suggestions from MJB, JC, JHZ, HBF, HT, IJ, NC, AB, TaB, MJ, SR, RRAB, AN, PTK and KF. IG constructed the search.

**Funding** MJB is supported by the Wellcome Trust (207472/Z/17/Z). JR is a Commonwealth Rutherford Fellow funded by the UK government through the Commonwealth Scholarship Commission in the UK. The Lancet Global Health Commission on Global Eye Health is supported by The Queen Elizabeth Diamond Jubilee Trust, Moorfields Eye Charity [grant number GR001061], NIHR Moorfields Biomedical Research Centre, The Wellcome Trust, Sightsavers International, The Fred Hollows Foundation, The SEVA Foundation, The British Council for the Prevention of Blindness and Christian Blind Mission.

**Competing interests** None declared.

**Patient and public involvement** Patients and/or the public were not involved in the design, or conduct, or reporting, or dissemination plans of this research.

**Patient consent for publication** Not required.

**Provenance and peer review** Not commissioned; externally peer reviewed.

### ORCID iDs
Ana Patricia Marques http://orcid.org/0000-0001-8242-7021
Jacqueline Ramke http://orcid.org/0000-0002-5764-1306
Justine H Zhang http://orcid.org/0000-0001-8385-2003
Serge Resnikoff http://orcid.org/0000-0002-5866-4446

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
