## [Reviewer comments · BMJ Open]

ARTICLE DETAILS

TITLE (PROVISIONAL)	Estimating the global cost of Vision Impairment and its major causes: protocol for a systematic review
AUTHORS	Marques, Ana Patricia; Ramke, Jacqueline; Cairns, John; Butt, Thomas; Zhang, Justine; Faal, Hannah; Taylor, Hugh; Jones, Iain; Congdon, Nathan; Bastawrous, Andrew; Braithwaite, Tasanee; Jovic, Marty; Resnikoff, Serge; Nandakumar, Allayala; Khaw, Peng; Bourne, Rupert; Gordon, Iris; Frick, Kevin; Burton, Matthew J

VERSION 1 – REVIEW

REVIEWER	David Keegan Mater Misericordiae University Hospital Ireland
REVIEW RETURNED	09-Feb-2020

GENERAL COMMENTS	This is a timely and needed study to gain greater understanding of economic (global and individual) impact of VI. The study as presented should attain those goals. The limitations are alluded to at the beginning of the paper but could be discussed more. They would include, Accurately identifying levels of sight preserving / saving therapy use in high, medium and low income countries. The ability to identify the range of therapeutic use will be challenging without access to electronic records (perhaps the discussions with individual authors of other studies will glean this information). Assumptions around costs of informal care do lead to inaccuracy.
--

REVIEWER	Alexis Malkin New England College of Optometry, USA
REVIEW RETURNED	20-Feb-2020

GENERAL COMMENTS	This is a well designed protocol that should produce a thorough systematic review. This review will add significantly to the existing literature looking at the costs associated with vision impairment. The authors are following a standard protocol and have explicitly described the process they will use for the review. I look forward to seeing the results.
--

REVIEWER	Antonio Filipe Teixeira Macedo Linnaeus University Sweden
REVIEW RETURNED	03-Mar-2020

GENERAL COMMENTS	General comments
------------------

	I read with great interest the above-mentioned manuscript. The manuscript reports the methodology for a systematic review looking at the costs of vision impairment worldwide. The motivations for conducting this systematic review are, amongst others, the lack of data from developing countries in previous reports and the need for updated information due to changes in treatment solutions worldwide in the past 2 decades. The manuscript is well written and structured. I have a few minor points. The topic is relevant and timely. Minor points Page 5, second paragraph -- some of the cited literature reporting consequences of eye diseases 7 vision loss is almost 20 years old. It would be interesting to have also more recently published studies. Page 6, paragraphs – the first 5 paragraphs are a mixture of background and methods. My best advice here, is to cut substantially in the methodological aspects because they are distracting – just keep the text straight to the justification for this new study. Some technical aspects can be moved to, for example, A) a new section in Methods or B) add them to the current section “Cost classification description”. If option B), maybe the heading of the section should be also updated. Page 11, first paragraph – there are a few transformations that the “raw” data will undergo to then be converted to US purchasing power parities. Whilst some of the concepts may be easy to grasp to experts, it may not be case for clinicians or other researchers interested in this report. I suggest to add a bit more detail about this methodology using, eventually, some examples.
--	--

REVIEWER	Debbie Muirhead The Fred Hollows Foundation and University of Melbourne, Melbourne, Australia
REVIEW RETURNED	02-Apr-2020

GENERAL COMMENTS	In its current form I would suggest the protocol does not read as having enough clarity three key areas below (some more detailed comments included in separate file) to be a thorough systematic review protocol nor to mitigate risk misinterpretation of results (a problem that plagues cost of illness studies and their use). Hence would suggest giving some more attention to these and resubmitting. a) structuring and representation of costs in the results – to guide bounds of what would be captured in or left out of various outcomes / summary measures (in this case cost categories) quantified in the review (by VI, disease or type of cost / economic impact). Whilst a meta-analysis is unlikely to be possible as the authors state due to the different ranges of types of costs and definitions used in the underlying studies, it is exactly this that makes such a structure / taxonomy – all the more important in a SR related to cost of illness, costing or economic impact studies. b) how quality of studies will be assessed and therefore validity of costs contained therein (referred to in PRISMA and still applicable here) c) how any outcome level of bias would be assessed and described for the summary measures to be used (e.g. for cost by VI which is stated as one of the potential aggregations desired) (again referred to in PRISMA and still relevant here). This
--

	important to consider items such as what and how productivity costs & intangible costs have been included, ensuring quality of life and productivity costs reported separately (considering potential or not for double counting), relevant inclusions and exclusions (e.g. excluding transfer payments if societal perspective used) etc. Note that currently some inclusions and exclusions either need a bit more clarity:  • on how some included costs would be used in what has been defined as a Col systematic review (e.g. (e.g. from CEA or CUA - just in direct health care cost component as treatment cost for relevant eye condition or would place a value on QALYs / DALYs reported and if so using what approach?) • on why some exclusion exist – e.g. seem to imply would not include studies employing modelling from other data – which is large number of Col studies (and some of the more generalisable – so long as they are clear on what they include, exclude and how they have valued key components like productivity costs). I have attached more detailed comments in a separate document in case this is useful to authors. Suggest some considerations outlined in Chisholm D, Stanciole Torres Edejer TT & Evans D Economic impact of disease and injury: counting what matters. BMJ 2010; 340 on what should be defined for such studies (and hence considered in SRs of them).
--	---

VERSION 1 – AUTHOR RESPONSE

Reviewer 1

This is a timely and needed study to gain greater understanding of economic (global and individual) impact of VI. The study as presented should attain those goals. The limitations are alluded to at the beginning of the paper but could be discussed more. They would include, accurately identifying levels of sight preserving / saving therapy use in high, medium and low income countries. The ability to identify the range of therapeutic use will be challenging without access to electronic records (perhaps the discussions with individual authors of other studies will glean this information). Assumptions around costs of informal care do lead to inaccuracy.

Response: Thank you for your feedback. We have expanded the study limitations outlined in the Strengths and limitations section. The text with study limitations (on page 4) has been amended as follows:

“Synthesis of findings will be difficult as resource use (including diagnostic procedures and treatment options) and costs will likely vary between countries, over time and according to which cause(s) of vision loss is reported— in lieu of synthesis we will summarise the range and quality of available evidence, and the subsequent gaps where evidence should be produced and improved.”

Reviewer 2

This is a well designed protocol that should produce a thorough systematic review. This review will add significantly to the existing literature looking at the costs associated with vision impairment. The authors are following a standard protocol and have explicitly described the process they will use for the review. I look forward to seeing the results.

Response: Thank you for this positive feedback.

Reviewer 3

General comments

I read with great interest the above-mentioned manuscript. The manuscript reports the methodology for a systematic review looking at the costs of vision impairment worldwide. The motivations for conducting this systematic review are, amongst others, the lack of data from developing countries in previous reports and the need for updated information due to changes in treatment solutions worldwide in the past 2 decades. The manuscript is well written and structured. I have a few minor points. The topic is relevant and timely.

Response: Thank you very much for your helpful comments and suggestions.

Minor points

Page 5, second paragraph -- some of the cited literature reporting consequences of eye diseases 7 vision loss is almost 20 years old. It would be interesting to have also more recently published studies.

Response: Thanks for this suggestion. References has been revised and more recent studies included, such as:

Black AA, Wood JM, Lovie-Kitchin JE. Inferior visual field reductions are associated with poorer functional status among older adults with glaucoma. *Ophthalmic Physiol Opt.* 2011 May;31(3):283-91

Zheng Y, Wu X, Lin X, Lin H. The prevalence of depression and depressive symptoms among eye disease patients: a systematic review and meta-analysis. *Sci Rep.* 2017 Apr 12;7:46453

Marques AP, Macedo AF, Hernandez-Moreno L, Ramos PL, Butt T, Rubin G, et al. The use of informal care by people with vision impairment. *PLOS ONE.* 2018;13(6):e0198631

Wang M-T, Ng K, Sheu S-J, Yeh W-S, Lo Y-W, Lee W-J. Analysis of Excess Direct Medical Costs of Vision Impairment in Taiwan. *Value in Health Regional Issues.* 2013;2(1):57-63

Page 6, paragraphs – the first 5 paragraphs are a mixture of background and methods. My best advice here, is to cut substantially in the methodological aspects because they are distracting – just keep the text straight to the justification for this new study. Some technical aspects can be moved to, for example, A) a new section in Methods or B) add them to the current section “Cost classification description”. If option B), maybe the heading of the section should be also updated.

Response: Thanks for pointing this out. We have reduced this explanation in the background and moved the material to a renamed subsection “Methodological features of cost of illness studies” in the Methods section (page 9).

Page 11, first paragraph – there are a few transformations that the “raw” data will undergo to then be converted to US purchasing power parities. Whilst some of the concepts may be easy to grasp to experts, it may not be case for clinicians or other researchers interested in this report. I suggest to add a bit more detail about this methodology using, eventually, some examples.

Response: We have revised and expanded the synthesis results study section to introduce your suggestion. The text with this description can be found on page 11 under Synthesis of results:

“Time transformations will adjust for inflation costs reported in the same country but in different years. Conversion to US dollar PPP will adjust for the same price level cost estimates reported in different countries and different currencies. This cost transformation will convert all reported costs to the same year (2018), same currency and same purchasing power (USD PPP)”

Reviewer 4

In its current form I would suggest the protocol **does not read as having enough clarity three key areas below** (some more detailed comments included in separate file) to be a thorough systematic review protocol nor to mitigate risk misinterpretation of results (a problem that plagues cost of illness studies and their use). Hence would suggest giving some more attention to these and resubmitting.

a) **structuring and representation of costs in the results** – to guide bounds of what would be captured in or left out of various outcomes / summary measures (in this case cost categories) quantified in the review (by VI, disease or type of cost / economic impact). Whilst a meta-analysis is unlikely to be possible as the authors state due to the different ranges of types of costs and definitions used in the underlying studies, it is exactly this that makes such a structure / taxonomy – all the more important in a SR related to cost of illness, costing or economic impact studies.

Response: Thank you for this point. We have expanded the Synthesis of Results section to address this. The text with this description (page 11) has been amended as follows:

“We will take four steps to prepare study results for comparison:

- 1) we will categorise studies as either ‘general’ studies that reported costs for people with blindness or VI or ‘condition-specific’ studies that reported costs for people with one of the seven specified causes of vision loss;*

- 2) *if costs per patient per year are not reported for national or global estimates studies, these will be calculated for studies where sufficient information is provided;*
- 3) *costs will be inflated to 2018 values (or to the recent available year) using country-specific GDP deflators³⁷; and*
- 4) *costs will be converted to USD purchasing power parities (PPP)³⁸ to equalise the purchasing power of different currencies.”*

To respond to a request made by reviewer 3 we have added an explanation about the costs transformation process and therefore after the four steps the following sentences has been added:

“Time transformations will adjust for inflation costs reported in the same country but in different years. Conversion to US dollar PPP will adjust for the same price level cost estimates reported in different countries and different currencies. This cost transformation will convert all reported costs to the same year (2018), same currency and same purchasing power (USD PPP)”

After this we continue our explanation about how we will synthesise results:

“Due to anticipated heterogeneity in the cost data, studies will be stratified and presented by the four different costs components (i.e. Direct Costs, Productivity losses, Informal care and Intangible costs), with a clear explanation of what is included in the four costs components. A table summarizing which items are included in the four major cost components will be reported to summarise the similarities and differences between studies.

Cost data will also be stratified by severity of VI when this information is available. Since this systematic review aims to collect data to assist a future global economic estimate of the cost of VI and its major causes, the transformed costs per patient per year stratified by costs components will be aggregated by GDB Region and Super Region. Descriptive statistics measures will be calculated to report the costs per patient per year for each GBD region and super-region (e.g. mean, standard deviation, minimum and maximum).”

b) how quality of studies will be assessed and therefore validity of costs contained therein
(referred to in PRISMA and still applicable here)

Response: Our plan for appraisal of studies is outlined in the ‘Selection of sources of evidence’ section. We can move this to its own section and add as a supplementary file the adapted checklist if the journal prefers. We have expanded the text of the ‘Selection of sources of evidence’ section to clarify how appraisal will be done. The text with this description (page 10) has been amended as follows:

“Each quality criterion will be scored as one of “yes,” “no,” “partial,” or “not applicable”. We will follow the approach used several times previously to identify the methodological strengths and weakness of the included studies^{32 35 36}—equal weight will be assigned to each item of the checklist and the final score will be equal to the sum of the 10 individual items.”

c) how any outcome level of bias would be assessed and described for the summary measures to be used (e.g. for cost by VI which is stated as one of the potential aggregations desired) (again referred to in PRISMA and still relevant here). This important to consider items such as what and how productivity costs & intangible costs have been included, ensuring quality of life and productivity costs reported separately (considering potential or not for double counting), relevant inclusions and exclusions (e.g. excluding transfer payments if societal perspective used) etc.

Response: Thank you for this useful suggestion. As outlined above, reports will be stratified by cost components to ensure that double counting is avoided. We will also include a table to summarise which items are included in the 4 cost components categories (Direct Costs, Productivity losses, Informal care and Intangible costs) and to report similarities and differences between studies that support and clarify which results are likely to be comparable and which are not. We have expanded the Synthesis of Results section to address your suggestion. The text with this description (page 11) has been amended as follows:

“A table summarizing which items are included in the four major cost components will be reported to summarise the similarities and differences between studies.”

Note that currently some inclusions and exclusions either need a bit more clarity:

- on how some included costs would be used in what has been defined as a Col systematic review (e.g. (e.g. from CEA or CUA - just in direct health care cost component as treatment cost for relevant eye condition or would place a value on QALYs / DALYs reported and if so using what approach?)

Response: Thanks for pointing this out. We have revised and expanded the Synthesis of Results section. The text with this description (page 12) has been amended as follows:

“Loss of well-being measures will be summarized in their natural units (eg. QALYS and DALYS) rather than reported in their monetized value since there is no consensus on assigning a monetary value to health outcomes^{21 26 39} and because there is no common acceptable value across countries”

- on why some exclusion exist – e.g. seem to imply would not include studies employing modelling from other data – which is large number of Col studies (and some of the more generalisable – so long as they are clear on what they include, exclude and how they have valued key components like productivity costs).

Response: Thanks for this suggestion which helped us to clarify our criteria. We included modelling studies that calculated costs either by collecting raw data or synthesising data from different sources and adding assumptions if needed. We excluded studies that did not report primary costs data or were based on reviews of existing economic studies. This has now been added to our eligibility criteria in the PICOS descriptive table (study design – page 8):

“Model based economic evaluation studies not reporting any primary cost data or based on reviews of existing economic studies were excluded”

I have attached more detailed comments in a separate document in case this is useful to authors.

Response: I was unable to locate this file, but would be happy to address any further comments if appropriate.

Suggest some considerations outlined in Chisholm D, Stanciole Torres Edejer TT & Evans D
Economic impact of disease and injury: counting what matters. BMJ 2010; 340 on what should be defined for such studies (and hence considered in SRs of them).

Response: Thank you for this suggestion. I have read this and incorporated into my response to this review where appropriate. In our systematic review we will make particular efforts to highlight where studies can and cannot be compared using the principles of this paper as outlined previously when we address your suggestion labelled as c).

VERSION 2 – REVIEW

REVIEWER	Antonio Filipe Macedo Linnaeus University Department of Medicine and Optometry Sweden
REVIEW RETURNED	17-May-2020
GENERAL COMMENTS	Thank you for correcting the manuscript. I am satisfied with the answers provided and the corrections implemented in the new version of the manuscript.